# Optimal Power Flow based on Area Partitioning Method for Power Grids

Zhiqiang Ma*, Fei Liang, Qi Yang, Bing Chen, Mei Zhou, Yu Na

*Metrology Centre, Marketing Service Center, State Grid Ningxia Electric Power Co., Yinchuan 750010, China*
e-mail: mf.512@163.com

*Abstract*—**The power system is gradually orienting to large-scale distributed computing, and the trend of area interconnection is becoming more and more obvious. It is still a classic problem to calculate power flow by using distributed optimization. Compared with traditional centralized power flow calculation, it has obvious advantages with fast computing speed and less information between areas. In this paper, we define the boundary voltage coupling equation with auxiliary variables, to solve the non-convex subproblem in the iterative process of ADMM, and match the optimal area partitioning for ADMM by proposing the optimal spectral clustering (OSC) method. Specifically, the OSC scheme takes the Jacobian matrix, admittance matrix, and voltage pivot vector as the the joint adjacency matrix to measure the similarity of spectral clustering algorithm into consideration, which is obviously better than the traditional spectral clustering method considering admittance matrix only. Combining with ADMM algorithm, IEEE-118 bus system is used as a test example to perform the distributed calculation after area partitioning, the experimental results verify the good performance of proposed spectral clustering method in distributed power grid.**

*Index Terms*—**Optimal power flow, spectral clustering, area partitioning, distributed power grid.**

## I. INTRODUCTION

In recent years, distributed optimization has received a lot of attention for solving problems that arise in power systems operations, as it provides a promising alternative for solving complex optimization problems associated with grids that have a large number of distributed generation units. This technique allows dividing an optimization problem into subproblems associated with different regions of the grid, which are solved separately and simultaneously with periodic information exchanges. One key application considered for distributed optimization is the Optimal Power Flow (OPF) problem, which is at the heart of power systems operations and planning.

Several practical issues arise when implementing distributed methods on large-scale networks. One critical issue is the system partitioning, which has been shown to have considerable impact on the performance of the distributed algorithm [1]. Spectral Clustering [2] is a widely used clustering algorithm, originally inspired by Graph Theory. Among them, the idea of undirected weight graph and cut can be well applied to the distribution network diagram of power system and the area partitioning idea for the OPF problem of distributed power grid.

In the traditional clustering algorithm, $K$-Means algorithm has been widely used, but it usually solves the convex problem,

the data is compact, and the initial cluster center is more sensitive, as shown in Fig. 1(a). Hence, this method does not address the important question of how to partition a general large-scale network in a way that is suitable for distributed optimization. In fact, due to the non-convexity introduced by the AC power flow equations, OPF is known to be a difficult problem to solve even in a centralized manner for large systems [3]. Further difficulties arise in the search for a distributed solution since optimality or even convergence cannot be guaranteed for most distributed methods on non-convex problems.

For non-convex problems, the data has continuity, and the use of spectral clustering has obvious advantages. When processing data with continuity in Fig. 1(b), spectral clustering mainly connected the data (that is, points in the space) through the idea of the graph, and the edges connected by each node had different weights, and the edges of its adjacent nodes had higher weights. The cutting idea is used to achieve the optimal weight of the subgraph after cutting. That is, the data weight of the boundary coupling is low, and the internal weight of the subgraph is high, completing the clustering.

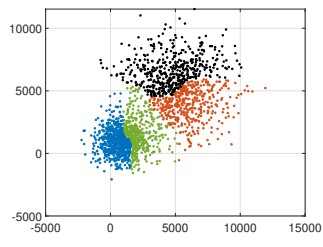 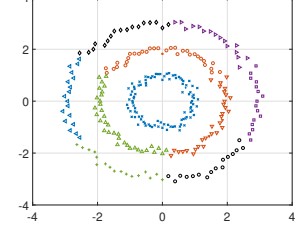

(a) $K$-means algorithm.      (b) Spectral clustering algorithm.

Fig. 1. Ablation study on the accuracy of different skip-edge rates.

In this paper, we proposed an area partitioning method based on optimal spectral clustering that significantly improves the convergence speed of distributed optimization. This partitioning method takes the Jacobian matrix, admittance matrix, and voltage pivot vector as the the joint adjacency matrix to measure the similarity of spectral clustering algorithm into consideration, which is obviously better than the traditional spectral clustering method considering admittance matrix only. Hence, our OSC method can not only capture the coupling among the areas, but also take the optimality condition of

OPF problem into consideration. Finally, the power system partitioning problem can be mapped to a graph partitioning problem. Combining with ADMM algorithm, IEEE-118 bus system is used as a test example to perform the distributed calculation after area partitioning, the experimental results verify the good performance of proposed spectral clustering method in distributed power grid.

## II. SPECTRAL CLUSTERING FOR AREA PARTITIONING

Spectral clustering method is proposed based on the graph models, which usually considers undirected graphs with weights. Define a graph $\mathcal{G} = (\mathcal{V}, \mathcal{E})$, where $\mathcal{V} = [v_1, \cdots, v_N]$ denotes the node set in the graph $\mathcal{G}$, and $\mathcal{E}$ denotes the edge set. For each edge, the weight matrix is defined as $\mathcal{W} = [w_{nn'}]$, $1 \leq n, n' \leq N$. $\mathcal{W}$ is called the similarity matrix (also called the adjacency matrix), and the different weight $w_{nn'}$ represents the connection relationship between nodes, which can be measured by defining the distance between nodes. Specially, when considering the undirected weight graph, $w_{nn'} = w_{n'n}$, as shown in Fig. 2.

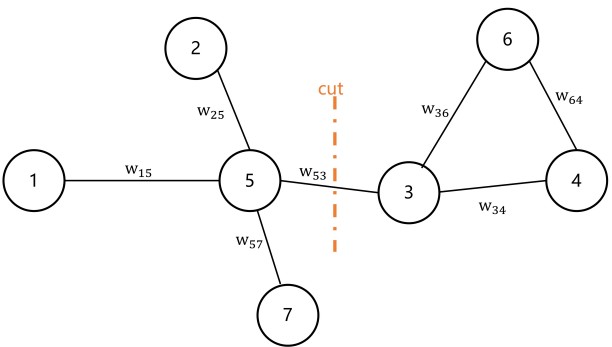

Fig. 2. The undirected weight graph.

Due to the different similarity between node pair, the weight set between each two adjacent nodes is greater than $0$, and the weight between other non-adjacent nodes is equal to $0$. The weight of each edge can be represented by a kernel function, commonly used can be Gaussian kernel RBF, as follows:

$$w_{nn'} = \begin{cases} 0, & (n, n') \notin \mathcal{E}, \\ \mathcal{K}(x_n, x_{n'}) = \exp\left(-\frac{\|x_n - x_{n'}\|_2^2}{2\sigma^2}\right), & (n, n') \in \mathcal{E}. \end{cases}$$

In order to utilize the idea of spectral clustering method to do area partitioning for optimal power flow, we need to partition the graph. Figure 2 defines the weight undirected graph $\mathcal{G} = (\mathcal{V}, \mathcal{E})$, we aim to divide it into $K$ subgraphs. In such $K$ subgraph, the node set in each subgraph are represented by $C_k \in [C_1, C_2, \ldots, C_K]$, that is $\mathcal{V} = \cup_{k=1}^{K} C_k$, $C_k \cap C_{k'} = \varnothing$. Define the objective as follows:

$$\text{cut}(\mathcal{V}) = \text{cut}(C_1, C_2, \ldots C_K) = \sum_{k=1}^{K} \mathcal{W}(C_k, \bar{C}_k) \quad (1)$$

where $\bar{C}_k$ denotes the other node sets except $C_k$.

Obviously, when the objective function minimizes the cut function $\mathcal{V}$, the minimized weights can be achieved. Due to the variable number of subgraphs, the minimization cut needs to be normalized, defined N-cut as follows

$$\text{N-Cut}(C_1, C_2, \ldots, C_k) = \sum_{k=1}^{K} \frac{\mathcal{W}(C_k, \bar{C}_k)}{\Delta_k} \quad (2)$$

where $\Delta_k$ denotes the 'degree' of the $k$-th subgraph. For the node $v_n$ in graph $\mathcal{G} = (\mathcal{V}, \mathcal{E})$, the degree defines the weight sum of all the edges adjacent to it, that is $d_n = \sum_{n=1}^{N} w_{nn'}$. Define the diagonal degree matrix for each node in graph $\mathcal{G}$, as follows:

$$\mathcal{D} = \begin{pmatrix} d_1 & \cdots & \cdots \\ \cdots & d_2 & \cdots \\ \vdots & \vdots & \ddots \\ \cdots & \cdots & d_N \end{pmatrix} \quad (3)$$

Further, the optimization objective (1) becomes

$$\text{N-Cut}(C_1, C_2, \ldots C_K) = \sum_{k=1}^{K} \frac{\mathcal{W}(C_k, \bar{C}_k)}{\sum_{n \in C_k} d_n}. \quad (4)$$

Note that solving (4) needs to define the Laplacian matrix $\mathcal{L}$. For the graph $\mathcal{G}$, the Laplacian matrix is $\mathcal{L} = \mathcal{D} - \mathcal{W}$. Define an indicator $h_n$ as

$$h_n = \begin{cases} 0, & v_n \notin C_k \\ \frac{1}{\sqrt{\Delta_k}}, & v_n \in C_k \end{cases} \quad (5)$$

which expresses the ownership relationship between the node and the area. Specifically, when the node does not belong to the current region, and $h_n$ is equal to zero. Collect all of $h_n$ into the matrix form of the subgraph $k$ as $\mathcal{H}$, and normalize it $\mathcal{H} = \mathcal{D}^{-1/2}\mathcal{U}$. Thus, the eigenmatrix $\mathcal{U}$ is obtained, and the optimization objective of spectral clustering is as follows:

$$\begin{array}{ll} \underset{\mathcal{U}}{\text{argmin}} & \text{tr}\left(\mathcal{U}^{\top}\mathcal{D}^{-1/2}\mathcal{L}\mathcal{D}^{-1/2}\mathcal{U}\right) \\ \text{subject to} & \mathcal{U}^{\top}\mathcal{U} = \mathcal{I} \end{array} \quad (6)$$

where $\mathcal{D}^{-1/2}\mathcal{L}\mathcal{D}^{-1/2}$ is the normalized Laplacian matrix $\mathcal{L}$, and 'tr' is the trace of the matrix. Cluster the eigenmatrix $\mathcal{U}$ by the $K$-means algorithm to get the area cluster partitioning.

## III. DISTRIBUTED MULTI-AREA OPTIMAL POWER FLOW

The optimal power flow (OPF) problem is a quadratic non-convex problem, in which the goal is to minimize the cost of electricity generation, the constraints are the power flow equation, the power flow inequality, the power limits and the voltage amplitude limits. Let $\mathcal{N} = [1, \ldots, N]$ be the node set of the large-scale power grid system, $(P_n, Q_n, V_n)$ be the triple of the active power, the reactive power and the voltage amplitude, and $(P_n^D, Q_n^D)$ be the tuple of the active and reactive power load. The system admittance matrix is $Y_{nn'}$,

and the set of bus nodes $n$ is $\Omega_n$. The standard problem for optimal power flow is as follows:

$$\underset{P,Q,V}{\text{minimize}} \quad f = \sum_{n=1}^{N} \left( \alpha_n P_n^2 + \gamma_n P_n + c_n \right) \tag{7}$$

$$\text{subject to} \quad P_n + jQ_n - P_n^D - jQ_n^D = V_n \sum_{n' \in \Omega_n} Y_{nn'}^* V_{n'}^*$$

$$P_n^{\min} \le P_n \le P_n^{\max}$$
$$Q_n^{\min} \le Q_n \le Q_n^{\max}$$
$$V_n^{\min} \le |V_n| \le V_n^{\max}$$

To implement the OPF problem (7) in a distributed manner, we need to partition the power system into smaller subareas and formulate a local OPF problem in each area. Then, the distributed OPF problem after decomposition is solved by an online iterative algorithm. Recall that $K$ denotes the total number of areas, and for $k = 1, \ldots, K$, $\mathcal{R}_k$ denotes the set of nodes assigned to the subarea $k$. The set $\mathcal{V}_k$ is introduced to denote the joint node set including both the nodes in $\mathcal{R}_k$ and the nodes in adjacent regions directly connected to nodes in $\mathcal{R}_k$, that is, $\mathcal{R}_k \subset \mathcal{V}_k$.

The distributed OPF problem is usually decoupled by replicating the voltage of the boundary bus in the adjacent areas. Constraints are then added so that the replicated voltages remain equal to each other, as shown in Fig. 3, where the connection line $nn'$ is the coupling branch connecting node $n$ and node $n'$. At the same time, the voltages on node $n$ and node $n'$ are equal. In order to establish the equality constraints, copy the voltages on the nodes and assign copies to the area $A$, represented by $V_{n,A}$ and $V_{n',A}$. Similarly, the voltage copies assigned by the region $B$ are represented by $V_{n,B}$ and $V_{n',B}$. In order to ensure the equivalence with the original OPF problem, we add the constraints $V_{n,A} = V_{n,B}$ and $V_{n',A} = V_{n',B}$ to the original optimization model.

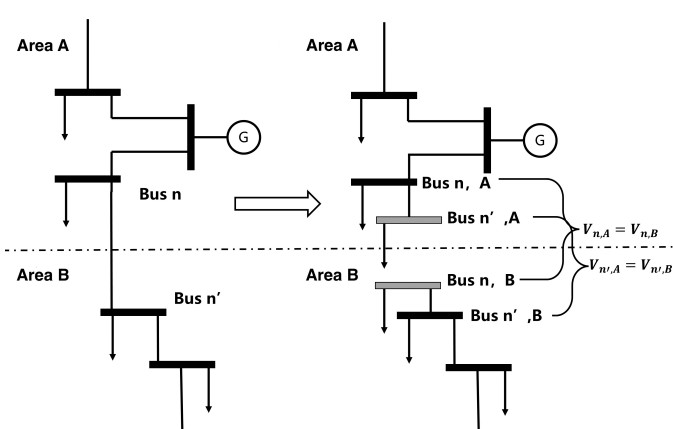

Fig. 3. The boundary coupling relationship between area A and area B and the replication constraint of voltage variables.

After replicating the voltage of the boundary bus in adjacent areas, we can cut the connection lines between adjacent areas, separating the adjacent area into separate subsystems, while

maintaining the original structure of the power grid by adding extra constraints. The constraints added are equivalent to

$$V_{n,A} - V_{n',A} = V_{n,B} - V_{n',B}$$
$$V_{n,A} + V_{n',A} = V_{n,B} + V_{n',B}. \tag{8}$$

For the branch $nn'$ satisfying $n \in \mathcal{R}_A$, $n' \in \mathcal{V}_A \backslash \mathcal{R}_A$, we regard $nn'$ as the coupling branch of area $A$. Further, introduce two auxiliary variable $z_{nn',A}$ and $z'_{nn',A}$ in area $A$ as the auxiliary constraints

$$z_{nn',A} = \eta \left( V_{n,A} - V_{n',A} \right)$$
$$z'_{nn',A} = \eta' \left( V_{n,A} + V_{n',A} \right) \tag{9}$$

where $\eta$ and $\eta'$ are scale factors relating closely to the current density through the coupling branch $nn'$. It is important to point out that the voltage difference between the branches is the explicit constraint for line coupling, hence $\eta$ should set greater than $\eta'$ to guarantee higher weight to $V_{n,A} - V_{n',A}$. Similarly, we also introduce two auxiliary variable $z_{nn',B}$ and $z'_{nn',B}$ into the area $B$. Thus, the feasible domain of all $z$ associated with the coupled branch in the area $B$ are defined as

$$z_{nn',B} = \eta \left( V_{n,B} - V_{n',B} \right)$$
$$z'_{nn',B} = \eta' \left( V_{n,B} + V_{n',B} \right) \tag{10}$$

Let $x_k = \{(V_n, P_n, Q_n) \mid n \in \mathcal{V}_k\}$ and $z_k = \{(z_{n,n'}, z'_{n,n'}) \mid n \in \mathcal{R}_k, n' \in \mathcal{V}_k \backslash \mathcal{R}_k\}$ denote all the primitive and auxiliary variables associated with bus nodes in the $k$-th area, respectively. Then the OPF problem (7) can be represented by the variables assigned to different areas, as follows:

$$\underset{x,z}{\text{minimize}} \quad \sum_{k=1}^{K} f_k\left(x_k\right)$$

$$\text{subject to} \quad A_k x_k = z_k, \quad \forall k$$
$$x_k \in \mathcal{X}_k, \quad \forall k$$
$$z_k \in \mathcal{Z}_k, \quad \forall k \tag{11}$$

where the set $\mathcal{Z} = \{(z^-, z^+) \mid z_{n,n'}^- = -z_{n,n'}^-, z_{n,n'}^+ = z_{n,n'}^+, \forall (n, n') \in \mathcal{E}\}$ is determined by the equality constraint (8) expressing via (10) and (9), and $\mathcal{E}$ denotes the branch set. In (11), $f_k\left(x_k\right)$ is the power generation costs of the $k$-th area, namely the objective function. By defining $x_k$ and $z_k$, the equation constraint of the problem and the feasible domain of variable $x_k$ are obtained, that is, the constraints of the variable $x_k$ for $\forall n \in \mathcal{R}_k$ and the constraints of the variable $z_k$ for $\forall n \in \mathcal{V}_k$. It is worth noting that for the OPF model (11), if $z_k$ is fixed which mean that the partition plan and further the information about boundary coupling branches are determined, then (11) can be decomposed into several sub-problems, each of which contains only the local variable $x_k$. Obviously, this problem can be solved using the distributed ADMM-based algorithms.

## IV. AREA PARTITIONING BASED ON OPF

The OPF problem of the large-scale power grids is very complicated and difficult to be solved directly, while the power system optimization problem is usually regional. Therefore, dividing the whole power grids into several sub-areas and

decomposing the OPF problem into several sub-problems can reduce the difficulty of solving the centralized OPF problem.

In this section, we simply partition the power grids based on a spectral clustering method proposed under optimal conditions to provide a distributed ADMM solution for power systems. Spectral clustering is a graph partitioning method that takes into account the similarity between any two nodes in the target graph, while in the OPF problem corresponds to the computational coupling between any two bus nodes. Specifically, the coupling relationship is reflected in the Jacobian matrix $\mathcal{H}$ and the admittance matrix $\mathcal{Y}$ and the voltage pivot vector $\Pi$. Finally, clustering is implemented by $K$-means algorithm to avoid the inaccuracy of clustering in non-convex areas. Let $K$ denote the number of sub-areas, and then find the optimal cluster that divides the system into these $K$ sub-areas. The entire procedure is as follows:

(1). Construct the Laplace matrix $\mathcal{L}$ including the Jacobian matrix $\mathcal{H}$ and the admittance matrix $\mathcal{Y}$. For the admittance matrix $\mathcal{Y}$, the non-diagonal incidence is represented by the negative modulus of the corresponding complex admittance, and the diagonal incidence is represented by the sum of the complex admittance coefficients, i.e.,

$$
\mathcal{Y}_{nn'} = \begin{cases} \sum_{n' \in \Omega_n} |y_{nn'}|, & n = n' \\ -|y_{nn'}|, & n \neq n', (n, n') \in \mathcal{E} \\ 0, & n \neq n', (n, n') \notin \mathcal{E} \end{cases}
$$

(2). Find the $K$ largest eigenvalues of $\mathcal{L}$, stack the corresponding eigenvectors $u_1, \cdots, u_K$ by columns and form the matrix $\mathcal{U} \in \mathbb{R}^{N \times K}$.

(3). Denote the $n$-th row vector of $\mathcal{U}$ by $v_n \in \mathbb{R}^K$, $n = 1, \cdots, N$, and cluster the point $\{v_n\}_{n=1}^N$ into the groups $C_1, \cdots, C_K$ by $K$-means algorithm.

(4). If the $n$-th row vector $v_n$ of $\mathcal{U}$ has been assigned to the cluster $C_k$, then the corresponding node $n$ will also be assigned to cluster $C_k$.

Finally, we discusses two issues when implementing the area partitioning method. First, $\mathcal{H}$ need be evaluated at the optimal position for a certain operating point, but if there is no drastic change in the power of the transmission line, using $\mathcal{H}$ evaluated by one of operating points are also suitable for solving OPF problems under the case of many operating points. For the robustness change to the system parameters such as generator cost coefficient after partitioning, if the change of parameters in the area is small, the partitioning plan is still applicable. On the other hand, if a significant or permanent change in the point of operation or system configuration occurs, the partitioning plan needs to readjust. Secondly, the bus in the same area obtained by the area partitioning method may not be connected physically to each other. Of course, only a small part of the actual distribution network (about 2%) is not connected. Moreover, even if the area is not completely connected, the subproblem can still be solved according to the definition.

## V. EXPERIMENTS

This section discusses the performance of the proposed area partitioning method for the OPF problem based on the classical ADMM algorithm [4]–[6], which is tested on the standard IEEE-118 bus system of MATLAB. The test is carried out on the basis of MATPOWER, using the Cplex package developed by IBM to solve the non-convex problem of local variables for the local OPF solution in the ADMM iteration. The initial Lagrange multiplier is set to 0, the initial penalty factor $\rho = 10^{-2}$, and other initial parameters $\eta$, $\eta'$ and $r$ are set to 0.5, 2, and 0.9, respectively. The stop condition for the iteration is that the maximum original residual is less than $10^{-4}$, i.e., ADMM relatively converges.

### A. Area Partitioning Performance

The conventional partitioning scheme can be done according to the electrical characteristics of the grid system, considering the relationship of line impedance between the generator and the load in the power grid. Previous research results [7], [8] show that the spectral clustering method has greater advantages than conventional partitioning schemes, so that won't be covered again here.

As one of the main differences, the classical spectral cluster based area partitioning scheme considers only the admittance matrix $\mathcal{Y}_{n,n'}$ as the benchmark adjacency matrix, denoted by SC($K$). The optimal spectral clustering scheme proposed in this paper utilizes the joint adjacency matrix composed of the Jacobian $\mathcal{H}_{n,n'}$, the admittance matrix $\mathcal{Y}_{n,n'}$ and the voltage pivot vector $\Pi$, denoted by OSC($K$). The total number of areas is denoted by $K$, and we take $K = 3$ in this experiment, and the partitioning results of two schemes are shown in Table I.

Table I
COUPLING PARAMETER AND AREA NODES FOR IEEE-118 BUS SYSTEM

| Method | Coupling Parameter | Area Nodes |
|---|---|---|
| OSC(3) | 0.7948 | 1-32,113-115,117;33-73,116;74-112,118 |
| SC(3) | 0.9107 | 1-7,11-15,33-37,39-64,117; 8-10,16-32,38,65-82,96-99,113-116,118; 83-95,100-112; |

From Table I, the coupling parameter of OSC(3) is obviously smaller than that of SC(3), that is, OSC(3) has achieved good results in terms of coupling parameters. Further one can conclude that the smaller the coupling parameter, the better the adaptability to distributed computing. In order to show the partitioning effect more intuitively, the experiment converts the data from the node diagram of the system into coordinate data, and calculates the cluster center according to the classification results, as shown in Fig. 4.

Table II analyzes the effect of two area partitioning methods on the number of iterations, running time, and the accuracy error between calculation result and optimal value. Obviously, the iteration number and running time of OSC will be smaller than that of SC, and the accuracy error obtained in limited time is higher. Although the conventional SC scheme measures the

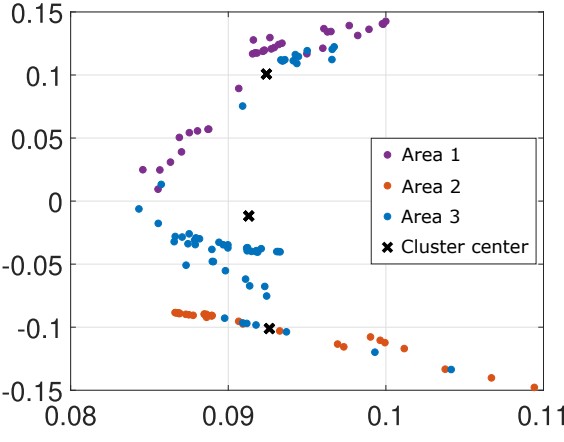

(a) Spectral clustering results of classical SC(3)

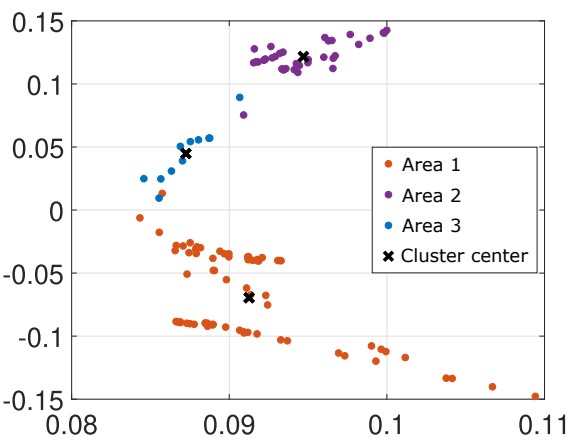

(b) Spectral clustering results of proposed OSC(3)

Fig. 4. Area node partitioning for IEEE-118 bus system when $K = 3$.

Table II
COMPUTING PERFORMANCE RESULT FOR IEEE-118 BUS SYSTEM

| Method | Iteration | Time (s) | Accuracy (%) |
|--------|-----------|----------|--------------|
| OSC(3) | 423 | 251.6 | 0.0503 |
| SC(3) | 675 | 326.8 | 1.130 |

line impedance among buses and the electrical distance between the motors through the admittance matrix, but the high imbalances phenomenon can still occur due to the complexity of the power grid. For example, there are several areas in SC(3) that contain loaded generators in the vicinity, which tend to be exacerbated on larger power grids. However, the OSC scheme not only ensures the balance of the optimal area partitioning, but also captures the Jacobian matrix $\mathcal{H}$ of the optimal conditions in the power grid, the admittance matrix $\mathcal{Y}$ with electrical characteristics and the voltage pivot vector, and comprehensively defines the adjacency matrix of the weights, so that it has more obvious advantages in large-scale power grids.

### B. Semi-supervised Node Classification

ADMM is the simplest and most effective distributed solution so far. Based on the advantages of spectral clustering, the convergence performance of ADMM is further evaluated in this subsection. As an iterative algorithm, ADMM's residual precision can well represent the performance of the algorithm. To this end, the original residuals and dual residuals of 3 areas on IEEE-118 bus system are compared respectively, and the OSC scheme is further compared with the conventional SC scheme.

This subsection further tests the ADMM convergence performance on IEEE-118 bus system, considering the primitive and dual residuals. As the topology complexity of the system increases, the iteration number of ADMM will increase significantly, which means that the performance gap between OSC and SC on ADMM may be further enlarged. Fig. 5 shows the convergence of primitive and dual residuals, where the running time of OSC(3) is 286.260920 seconds, and the running time of SC(3) is 359.307797 seconds. It is easy to see that the ADMM algorithm has a longer oscillation period, and the effect of initial convergence is not obvious. However, in each subproblems for the corresponding area, the OSC scheme shows obvious advantages, and the oscillation amplitude is much smaller than that of conventional SC scheme. In addition, the OSC scheme is significantly better than the SC scheme in terms of the algorithm stop time.

The conventional SC scheme only considers the admittance matrix $\mathcal{Y}$ on the line, which is difficult to take into account the electrical characteristics and minimize coupling parameter. Therefore, the OSC scheme proposed in this paper can often screen out the optimal partitioning method after balancing the optimal conditions.

## VI. CONCLUSION

The spectral clustering algorithm is essentially a graph-based clustering method, which has a good clustering effect on continuous non-convex datasets. The connection mode of the power system satisfies the characteristics of undirected graph, and there are different admittance weights on each branch. At the same time, as one of the best solutions to large-scale distributed problems, ADMM also has a lot of applications and research for multi-area interconnected power systems. By defining the combination of Jacobian matrix, admittance matrix and voltage pivot vector as similarity measure, not only the electrical characteristics of the system but also the optimality condition of OPF problem solving are considered, and the adaptation of region division to ADMM algorithm is strengthened. Using voltage as the decomposition equation between regions, the non-convex problems in the subproblems are eliminated, and the experimental simulation is carried out by IEEE-118 bus system. The results show that the spectral clustering algorithm based on optimality conditions and the distributed solution of ADMM are effective, and they are more significant in large-scale power grids.

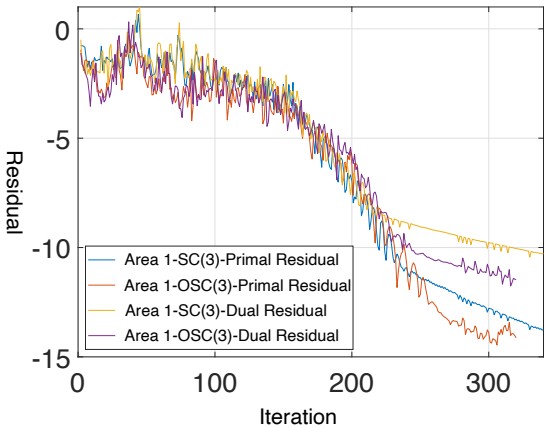

(a) Residuals of area 1.

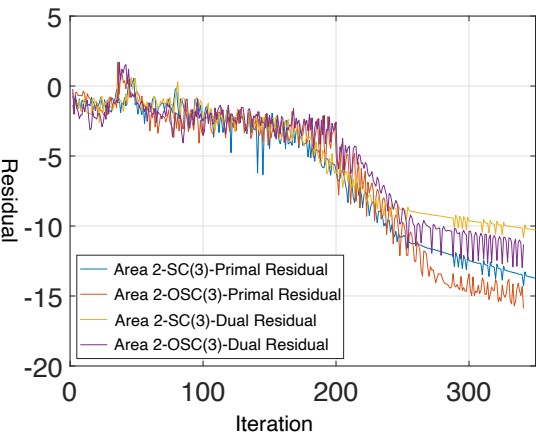

(b) Residuals of area 2.

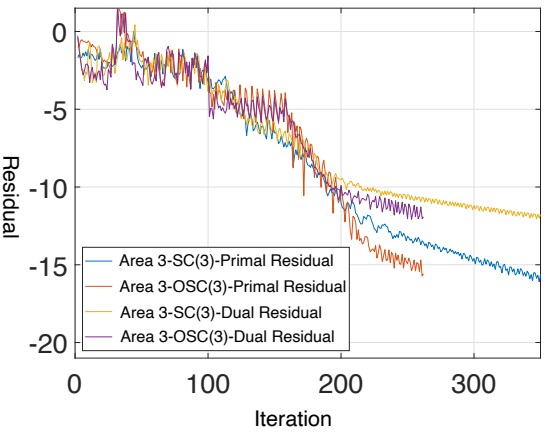

(c) Residuals of area 3.

Fig. 5. Convergence of original and dual residuals for IEEE-118 bus system.

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
