# OpenReview forum: "Optimal Power Flow based on Area Partitioning Method for Power Grids"
_IEEE.org/ICIST/2024/Conference — IEEE ICIST 2024 Conference Submission_

### Official Review · Reviewer_8FWs · 2024-08-21
**This article is quite fascinating and of high quality.**

**Rating:** 7
**Confidence:** 3

**Review:**

The paper titled "Optimal Power Flow based on Area Partitioning Method for Power Grids" proposes an optimal spectral clustering method. Firstly, a pseudo-label generator is proposed to label the training samples, which utilizes the Jacobian matrix, admittance matrix, and voltage principal vector as a combined adjacency matrix to assess the similarity of the spectral clustering algorithm. My specific feedback is as follows: 1) The research challenges and innovations of this paper can be further explained in this paper. Please add these details. 2) Some formatting issues need to be addressed.

---

### Official Review · Reviewer_83Th · 2024-08-21
**Accept**

**Rating:** 7
**Confidence:** 3

**Review:**

This paper defined the boundary voltage coupling equation with auxiliary variables, to solve the non-convex subproblem in the iterative process of ADMM, and matched the optimal area partitioning for ADMM by proposing the optimal spectral clustering (OSC) method. The theory is correct and can be accepted after responding the following comments.
(1)	In the introduction, it is not enough to state the current work. It should be expended and reconstructed.
(2)	There are many typos and grammar errors. The authors should have a native English speaker or software packages to perform the editing check.
(3)	The conclusion of the article suggests using the present perfect tense for description.

---

### Official Review · Reviewer_o2YV · 2024-08-28
**Optimal Power Flow based on Area Partitioning Method for Power Grids**

**Rating:** 7
**Confidence:** 2

**Review:**

In this paper, a graph-based clustering method is proposed, which is called the spectral clustering algorithm
a The references should be updated. Some closely related and new references should be added to show to further explain the novelty and innovation of the work.
b For the results presented in the Figures in the simulation, more explanations on them seem necessary and helpful to readers.

---

### Decision · Program_Chairs · 2024-09-06

Accept (Oral)